# *PyuARF16/33* Are Involved in the Regulation of Lignin Synthesis and Rapid Growth in *Populus yunnanensis*

**DOI:** 10.3390/genes14020278

**Published:** 2023-01-21

**Authors:** Zhixu Hu, Dan Zong, Qin Zhang, Xiaolin Zhang, Yu Lu, Chengzhong He

**Affiliations:** 1Key Laboratory for Forest Genetic and Tree Improvement &Propagation in Universities of Yunnan Province, Southwest Forestry University, Kunming 650224, China; 2Key Laboratory of State Forestry Administration on Biodiversity Conservation in Southwest China, Southwest Forestry University, Kunming 650224, China; 3Yunnan Academy of Biodiversity, Southwest Forestry University, Kunming 650224, China; 4Key Laboratory for Forest Resources Conservation and Use in the Southwest Mountains of China, Ministry of Education, Southwest Forestry University, Kunming 650224, China

**Keywords:** *ARFs*, red light, lignin, accelerated growth, *Populus yunnanensis*

## Abstract

(1) Background: Lignin is a unique component of the secondary cell wall, which provides structural support for perennial woody plants. *ARFs* are the core factors of the auxin-signaling pathway, which plays an important role in promoting plant growth, but the specific relationship between auxin response factors *(ARFs)* and lignin has not been fully elucidated with regard to rapid plant growth in forest trees. (2) Objectives: This study aimed to investigate the relationship between *ARFs* and lignin with regard to rapid plant growth in forest trees. (3) Methods: We used bioinformatics analysis to investigate the *PyuARF* family, find genes homologous to *ARF6* and *ARF8* in *Populus yunnanensis*, and explore the changes in gene expression and lignin content under light treatment. (4) Results: We identified and characterized 35 *PyuARFs* based on chromosome-level genome data from *P. yunnanensis*. In total, we identified 92 *ARF* genes in *P. yunnanensis*, *Arabidopsis thaliana*, and *Populus trichocarpa*, which were subsequently divided into three subgroups based on phylogenetic analysis and classified the conserved exon–intron structures and motif compositions of the *ARFs* into the same subgroups. Collinearity analysis suggested that segmental duplication and whole-genome duplication events were majorly responsible for the expansion of the *PyuARF* family, and the analysis of Ka/Ks indicated that the majority of the duplicated *PyuARFs* underwent purifying selection. The analysis of cis-acting elements showed that *PyuARFs* were sensitive to light, plant hormones, and stress. We analyzed the tissue-specific transcription profiles of *PyuARFs* with transcriptional activation function and the transcription profiles of *PyuARFs* with high expression under light in the stem. We also measured the lignin content under light treatment. The data showed that the lignin content was lower, and the gene transcription profiles were more limited under red light than under white light on days 1, 7, and 14 of the light treatments. The results suggest that *PyuARF16/33* may be involved in the regulation of lignin synthesis, thereby promoting the rapid growth of *P. yunnanensis*. (5) Conclusions: Collectively, this study firstly reports that *PyuARF16/33* may be involved in the regulation of lignin synthesis and in promoting the rapid growth in *P. yunnanensis.*

## 1. Introduction

Rapid growth is a complex quantitative trait controlled by external signals and endogenous signals, such as light [1]. Tree height and stem diameter are the most important traits associated with plants’ rapid growth [2]. The tree diameter is mainly determined by the secondary cell wall containing lignin, cellulose and hemicellulose and composed of vessels and fibers [3]. Lignin is the main structural component of plants’ secondary cell wall [4]. Although the lignin biosynthesis pathway was studied previously in woody plants [5,6,7,8], the relationship between lignin biosynthesis and tree height in woody plants remains unclear.

Auxin is one of the most widely found and studied plant hormones. Auxin signals play an important role in plant growth and development, including vascular tissue differentiation, root and stem morphogenesis, plant flowering, diurnal opening and closure [9,10], phototropism, apical dominance [11,12], development of secondary xylem [13,14].

*Populus yunnanensis Dode* is an important native tree species in China, characterized by rapid growth, excellent wood quality, strong adaptability, easy survival of the cuttings, high biomass, and environmental friendliness. It plays an important role in forestry production, ecological restoration, and environmental protection, and as a perennial model for forestry and biological research [15]. However, as far as we know, the *ARF* gene family in *P. yunnanensis* has not been studied. It would be interesting to supplement our understanding of *ARF* transcription factors in woody plants and study the relationship between *ARFs* and lignin synthesis in *P. yunnanensis* in order to provide a molecular basis for rapid growth that could be useful in tree breeding. The complete genome sequencing of *P. yunnanensis* has laid a good foundation for molecular biology research in *P. yunnanensis*. In this study, cuttings of *P. yunnanensis* exhibited rapid growth under red light irradiation, and this prompted us to investigate the involvement of auxin signaling pathways in lignin synthesis. We used bioinformatics-related techniques to identify the *ARF* family members in *P. yunnanensis* and analyze their structure and evolution, providing a certain reference for the study of growth factor-related genes in *P. yunnanensis*. We aimed to identify *PyuARF* genes with transcriptional activation function possibly involved in the regulation of lignin synthesis, promoting the rapid growth in *P. yunnanensis*.

This study speculated that an auxin signaling pathway is involved in the synthesis of lignin and promotes the elongation of the stem in *P. yunnanensis*. It provides a good theoretical basis suggesting that an auxin signal pathway regulates the thickness of the secondary cell wall to balance plant growth and an important foundation for further studies on the molecular mechanisms and genetic regulation of lignin biosynthesis and accelerated growth in forest trees.

## 2. Materials and Methods

### 2.1. Plant Materials and Treatment Conditions

The plant material was grown for one year in a greenhouse (Southwest Forestry University, Kunming, China). We obtained cuttings of about 15 cm and preserved the plants in the soil. The cuttings with 5–6 leaves were used in experiments performed in March 2022. The cuttings with uniform growth were placed in a constant-temperature incubator (temperature: 25 °C, humidity: 70%, light intensity: 90 µmol·m^−2^·s^−1^, photoperiod: 12 h light/12 h dark) for cultivation under red and white light (Telipu, Beijing Zhongyi Boteng Technology Company, power: 8 W, radiation intensity: 68 µW/cm^2^, wavelength: 620–760 nm, 380–760 nm) from lamps placed 35 ± 1 cm above the cuttings. For the quantification of gene expression by quantitative reverse transcriptase-PCR (RT-qPCR) and the measurement of lignin content, we collected the cuttings of *P. yunnanensis* after 0, 1, 7, and 14 days under red and white irradiation. We obtained samples from point A (1 cm from stem tip) to point B (1 cm from the base of the stem) of the stem and tissue samples without treatment of the roots, RAM (root apical meristem), stem, SAM (shoot apical meristem), young leaves (1st to 3rd leaves) and old leaves (7th to 9th leaves) [16], preparing three biological replicates for each sample. All the plant samples were frozen in liquid nitrogen and stored at −80 °C.

### 2.2. Identification and Sequence Analysis of ARFs in P. yunnanensis

The genome of *P. yunnanensis* was obtained from our laboratory group (Southwest Forestry University, Kunming, China). The hidden Markov model (HMM) file of the *ARF* family was extracted from the Pfam database (http://pfam.wustl.edu, accessed on 5 December 2021) [16], and SPDE was used to compare data from the protein database of *P. yunnanensis* (accessed on 5 December 2021) [17]. Then, the protein sequences of the *ARF* gene family members from *A. thaliana* and *P. trichocarpa* were obtained from the PlantTFDB database (http://planttfdb.gao-lab.org/, accessed on 5 December 2021) [18]. Based on the genome and GFF files of *P. yunnanensis*, we used TBtools to extract the proteome files of *P. yunnanensis* [19]. With Blastp, we compared the protein sequences of the *ARF* gene family members from *A. thaliana* and *P. trichocarpa* with those in the protein database of *P. yunnanensis* obtained by SPDE (accessed on 6 December 2021). Using the NCBI database (https://blast.ncbi.nlm.nih.gov/, accessed on 6 December 2021), we compared the data with those from the Uniprot protein database, to eliminate redundancy and predict the sequences of the *ARF* gene family [20]. We used SMART (http://smart.embl-heidelberg.de, accessed on 7 December 2021) and NCBI (CD-Search) (https://www.ncbi.nlm.nih.gov/Structure/bwrpsb/bwrpsb.cgi, accessed on 7 December 2021) to identify protein domains in the *ARF* family members. Finally, the existence of the *ARF* gene family in *P. yunnanensis* genome was determined. Using ProtParam (http://web.expasy.org/protparam/, accessed on 7 December 2021) [21], the analysis the *ARF* genes provided other information, including the number of amino acids, the molecular weight, and the isoelectric point of the ARF proteins [22].

### 2.3. Comparative Phylogenetic Analysis of ARFs

Multiple sequence alignments of the ARF protein sequences from *P. trichocarpa*, *A. thaliana*, and *P. yunnanensis* were performed using molecular evolutionary genetics analysis (MEGA) version.7 0 [23]. We used the Maximum Likelihood (ML) method to construct the evolutionary tree. Bootstrap was set to 1000 [24]. Subsequently, using the Interactive Tree of Life (iTOL) (https://pubmlst.org/, accessed on 8 December 2021), we visualized the ML phylogenetic tree [25]. Lastly, using the Tbtools software, we constructed a combination of phylogenetic tree, conserved domains, gene structures, and conserved motifs of *PyuARFs*.

### 2.4. Localization and Gene Duplication of ARFs

To map the physical positions of the identified *ARF* genes to the 16 chromosomes of the *P. yunnanensis* genom, we used the Tbtools software. The orthologous *ARF* genes were included in a local database using DIAMOND [26]. The *P. trichocarpa* and *A. thaliana* genomes (version = 3.0) were derived from the NCBI database. Using the Multiple Collinearity Scan toolkit (MCScanX) [27], we analyzed the gene duplication events with default parameters. To estimate the Ks (synonymous substitution rate) and Ka (nonsynonymous substitution rate) of collinearity pairs of the *ARF* gene family members in *P. yunnanensis*, we used the Tbtools software.

### 2.5. Prediction of Cis-regulatory Elements in the Promoters of PyuARFs

The promoters‘ upstream region (~2000 bp) sequences of *PyuARFs* were extracted from the genomic DNA sequence of *P. yunnanensis* and, subsequently, were submitted to the PlantCARE website (http://bioinformatics.psb.ugent.be/webtools/plantcare/html/, accessed on 9 December 2021) [28] for cis-acting element prediction. The results were analyzed by TBtools.

### 2.6. RNA Isolation and cDNA Synthesis

According to the relative manual, total RNA was isolated using the E.Z.N.A^@^ Plant RNA kit (Omega Bio-tek Inc., Norcross, GA, USA). We detected the quality and purity of RNA using K5800C (KAIAO, Beijing, China). According to the manufacturer’s protocols, 0.5 ug RNA of each sample was used for 1st strand cDNA synthesis using Hifair^®^ III Reverse Transcriptase (YEASEN, Shanghai, China). Subsequently, the cDNA was diluted 7-fold for RT-qPCR analysis.

### 2.7. Quantitative Real-Time PCR for the Evaluation of Gene Transcription Profiles

We determined the tissue-specific transcription profiles of *PyuARFs* and the transcription profiles of *PyuARFs* in the stems under light. We designed the primers for real-time PCR based on the CDS sequences using NCBI primer BLAST (Appendix A). The internal reference gene was HIS (histone) [29]. QRT-PCR was executed with the Hieff UNICON^®^ Universal Blue qPCR SYBR Green Master Mix (YEASEN, Shanghai, China) using a LightCycler^®^ 96 Real-Time PCR Detection System (Roche, Hercules, Switzerland). A total volume of 20 μL contained 10 μL of Blue qPCR SYBR Green Master Mix, 1 μL of cDNA samples, 0.4 μL of each primer (1 μM), and 8.2 μL of ddH_2_O. The PCR thermal cycle conditions were: 95 °C for 2 min, 45 cycles at 95 °C for 10 s and at 56 °C for 30 s. The melting curve conditions were: 1 cycle at 95 °C for 10 s, then at 65 °C for 10 s and 97 °C for 1 s. The relative gene expression was calculated using the 2^−ΔΔCt^ method [30]. The experiments were performed in triplicate technological repeats.

### 2.8. Measurement of Growth Height

The stem length from point A to point B was measured using a straight ruler for 14 days [17].

### 2.9. Measurement of Lignin Content

Cuttings spanning from point A to point B were dried, crushed it into powder, and passed through a 30–50-mesh sieve before component analysis. According to the manufacturer’s protocols, a 5 mg sample was weighed, placed in a test tube, and treated with 0.5 mL of reagent and 20 μL of perchloric acid, to obtain a blank sample. It was then sealed with a sealing film and mixed thoroughly. The samples were placed in a water bath at 80 °C for 40 min, shaken once every 10 min, and then cooled naturally. Next, we added 0.5 mL of reagent II in the tubes, fully mixed, and centrifuged the tubes at room temperature at 8000× *g* for 10 min. We then placed 10 μL of supernatant in tubes and added 0.99 mL of glacial acetic acid. After fully mixing, the absorbance was measured at 280 nm (Sangon Biotech, Shanghai, China). We named the samples A determination tube and A blank tube; therefore, ΔA = A determination tube -A blank tube was calculated, as reported [31].
Lignin content (mg/g) = ΔA/ε × V_1_/(V_2_ × W/V_3_) = 2.184 × ΔA/W. 

Here, V_1_ is the detection reaction volume, 1 mL, V_2_ is the liquid supernatant volume, 0.02 mL, V_3_ is the acetylation reaction volume, 1.02 mL, ε is the extinction coefficient of lignin, 23.35 mL/mg/cm. W is the sample quantity, g; 1000: conversion factor, 1 g = 1000 mg.

### 2.10. Data Analysis

Image processing was performed using Adobe Photoshop CS6, chart layout using Adobe Illustrator 2020, and data processing using Graphpad Prism 8.0.2 software.

## 3. Results

### 3.1. Genome-Wide Identification and Sequence Analysis of PyuARFs

Thirty-five *ARF* family genes were identified and named *PyuARF1–PyuARF35* according to their position on chromosomes (Table 1). They had different sequence lengths, resulting in different isoelectric points and molecular weights. The lengths in amino acids ranged from 391 aa (*PyuARF2*) to 1130 aa (*PyuARF35*), and the open reading frames ranged from 1176 bp (*PyuARF2*) to 3393 bp (*PyuARF35*). The molecular weight ranged from 43.46 kDa (*PyuARF2*) to 125.45 kDa (*PyuARF35*), and the isoelectric point ranged from 5.28 (*PyuARF17*) to 8.99 (*PyuARF24*). The number of exons was from 2 (*PyuARF6*) to 16 (*PyuARF11*).

### 3.2. Comparative Phylogenetic Analysis of the ARFs

In order to understand the evolutionary relationship of the *ARFs* from *A. thaliana*, *P. trichocarpa*, and *P. yunnanensis* and the differences in the *ARF* genes among different species, we constructed a rootless phylogenetic tree using the protein sequences of the *ARF* family members, i.e., those of 22 *AtARFs* from *A. thaliana*, 35 *PtrARFs* from *P. trichocarpa*, and 35 *PyuARFs* from *P. yunnanensis* (Figure 1).

### 3.3. Conserved Motif Domain and Structural Analysis of PyuARFs

We analyzed the exons and introns of the *PyuARFs* to determine the structural characteristics of the genes (Figure 2). The genes in the Class I family have fewer exons and introns, while others have more exons and introns. In addition, the length of the exons of *PyuARFs* increased from the 5′-terminus to the 3′-terminus. We analyzed the protein domains of *PyuARFs* to investigate the genes’ functions. All *PyuARFs* contain B3 domains and MR domains (Appendix A). In addition to the Class I and Class II features, the *ARF* (*PyuARF2*, *PyuARF29*, *PyuARF15*, *PyuARF31*, *PyuARF1*, *PyuARF21* and *PyuARF27*) genes contain *AUX/IAA* binding domains. The MR domain of the *ARF (PyuARF35*, *PyuARF34*, *PyuARF33*, *PyuARF27*, *PyuARF21*, *PyuARF18*, *PyuARF17*, *PyuARF16*, *PyuARF13*, *PyuARF5*, *PyuARF4* and *PyuARF3)* genes contains a high proportion of Q (glutamine), S (serine), and L (leucine) (Appendix A). We focused on the motifs of the *ARF* genes to analyze the functional composition of the gene domains. There are three major types of *ARF* genes in *P. yunnanensis*, in which each type of motif has high similarity and is conserved. All genes contain at least 10 motifs (Appendix A).

### 3.4. Localization and Duplication of PyuARFs

The high sequence similarity between repetitive gene pairs indicated that they were likely involved in regulating similar biological processes. We used MCScanX to analyze the collinear blocks and gene duplication type (Figure 3). Twenty-seven duplicated *PyuARF* pairs were observed in the *P. yunnanensis* genome. The duplication events, including segmental and whole-genome duplications during the evolutionary process of the *PyuARFs* family, were analyzed. The twenty-seven duplicated gene pairs appeared located on different chromosomes; except for chromosomes 7, 13, 19, all other chromosomes appeared to contain *PyuARFs* family members, and chromosomes 1, 2, 3, 4, and 6 appeared to contain the majority of *PyuARFs* (Figure 4). To gain a detailed understanding of the evolutionary constraints on *PyuARFs* family (Figure 5), the Ka/Ks ratios of the *PyuARF* pairs were analyzed (Figure 6).

### 3.5. Prediction of Cis-Regulatory Elements in the Promoters of PyuARFs

We then studied the potential regulation of cis-acting elements in *PyuARFs* expression. We analyzed 2 kb promoter sequences of *PyuARFs*. We found that MeJA (CGTCA motif), light-responsive (Box4), ABA (ABRE) anaerobic (ARE), gibberellin (GARE motif) motifs and response elements were abundant in the promoters of *PyuARFs*; salicylic acid (TCA motif), MYB drought (MBS), auxin (AuxRR core), low temperature (LTR), endosperm and meristem (GCN4), and circadian response elements were also found in the promoters (Figure 7).

### 3.6. Transcription Profiles Analysis of the ARF Genes in P. yunnanensis Organs

The 12 *PyuARFs* (*PyuARF3*, *PyuARF4*, *PyuARF5*, *PyuARF13*, *PyuARF16*, *PyuARF17*, *PyuARF18*, *PyuARF21*, *PyuARF27*, *PyuARF33*, *PyuARF34* and *PyuARF35*) showed tissue-specific transcription profiles in different organs in *P. yunnanensis*. In this experiment, the *PyuARFs* with high expression in the stem were selected for the next experiment; they included *PyuARF34*, *PyuARF33*, *PyuARF18*, *PyuARF16*, and *PyuARF5* (Figure 8).

### 3.7. PyuARFs Transcription Profiles under Light Irradiation

The transcription profiles of the five selected *PyuARFs* were examined under red light irradiation. On day 1, the transcription profiles of *PyuARF16/33* under red light were less rich than under white light. Similarly, on day 7 and on day 14, the transcription profiles of *PyuARF16/33* under red light were more limited than under white light (Figure 9).

### 3.8. Measurement of the Lignin Content

Lignin plays an important role in the rapid growth of plants. We measured the content of lignin in the cuttings of *P. yunnanensis* under light and observed the changes of lignin content on days 1, 7, and 14. On day 1, the lignin content under red light was lower than under white light. Similarly, on day 7 and on day 14, the lignin content under red light was lower than under white light (Figure 10).

### 3.9. Measurement of Stem Length

We explored the effect of the different types of light on the growth of the stem in *P. yunnanensis*. We measured the length of the stems at the beginning of the irradiation and after 14 days of irradiation (Figure 11).

## 4. Discussion

### 4.1. Identification and Analysis of PyuARFs Gene Family Members in P. yunnanensis

In this study, 35 *ARF* gene family members were identified from the genome of *P. yunnanensis*. We found that the PyuARF proteins vary greatly in both sequence length and protein characteristics, indicating that they have different characteristics. It is noteworthy that the isoelectric point of most ARF proteins from *P. yunnanensis* is less than 7, indicating that most *PyuARFs* may encode weakly acidic proteins and play biological functions in the acidic subcellular environment [32].

The 92 *ARF* family genes were divided into three categories (Class I, Class II and Class III). Class III included two subcategories, IIIa and IIIb. The MR domain of the *ARF* gene family of Class IIIb mainly contains Q (glutamine) and presumedly has a transcriptional activation function [33]. The MR domain of the *ARF* genes of class IIIb mainly contains SPL (serine, proline, and leucine) (Appendix A), and presumedly has transcriptional inhibition activity. Class Ia and Class II members have uncertain functions.

Intermediate structural regions (MR) are unique domains found in the *ARF* transcription factors family. They activate or inhibit target genes depending on their amino acid composition [34,35,36,37]. The C-terminal is a dimeric domain (CTD) consisting of the motifs III and IV of *AUX/IAAs* transcription factors, which can bind to *AUX/IAAs* factors to form heterodimers. When auxin concentrations are low, *AUX/IAAs* bind directly to *ARFs* to form heterodimers [38]. The *ARF* (*PyuARF2*, *PyuARF29*, *PyuARF15*, *PyuARF31*, *PyuARF1*, *PyuARF21* and *PyuARF27*) genes could bind *AUX/IAAs* in *P. yunnanensis*. The *ARF* (*PyuARF35*, *PyuARF34*, *PyuARF33*, *PyuARF27*, *PyuARF21*, *PyuARF18*, *PyuARF17*, *PyuARF16*, *PyuARF13*, *PyuARF5*, *PyuARF4* and *PyuARF3*) genes have the function of transcriptional activation.

We found Ka/Ks < 1 for 108 duplicated *ARF* gene pairs within *A. thaliana*, *P. trichocarpa*, and *P. yunnanensis* and Ka/Ks < 1 for 25 duplicated *ARF* gene pairs in *P. yunnanensis*, We speculate that *PyuARFs* undergwent selective pressure during evolution. Thus, we hypothesized that these gene duplication events led to a higher functional diversity in the *PyuARF* family members.

The promoter regions of 35 *PyuARF* genes also contain various cis-acting elements, such as MeJA (CGTCA motif) Light (Box4), ABA(ABRE) anaerobic (ARE), gibberellin (GARE motif), salicylic acid (TCA motif), MYB drought (MBS), low temperature (LTR), endosperm and meristem (GCN4), and circadian response elements. These results indicate that *ARFs* are involved in a variety of growth and development processes and in signal regulation in plants. Their functions in *P. yunnanensis* deserve further study.

### 4.2. PyuARF16/33 Are Involved in Lignin Biosynthesis

Red light plays an important role in photosynthesis and in the deposition of secondary cell walls [39,40]. The stem of *P. yunnanensis* showed a significant difference in height under red and white light. We found that different types of light had different effects on the growth and development of the stem in *P. yunnanensis*. The results showed that red light promotes stem elongation more than white light.

Previous studies reported that several transcription factors and hormones can regulate lignin biosynthesis and inhibit plant height by increasing lignin content [41,42]. Lignin can enhance cell wall strength and stem lodging resistance in soybean [43,44]. However, whether *ARFs* affect lignin synthesis to regulate the growth and development of the plant stem is still unclear. *ARF6* and *ARF8* show functional redundancy [45], because they regulate the development of secondary xylem in wood and hypocotyl elongation in *A. thaliana* [1,13]. According to homology alignment, *ARF6* and *ARF8* of *A. thaliana* match to Class IIb *ARFs* in *P. yunnanensis*. So, 12 *PyuARFs* were selected to analyze their transcription profiles and investigate their physiological functions according to their transcriptional activation in various tissues. Five *PyuARFs* (*PyuARF5*, *PyuARF16*, *PyuARF18*, *PyuARF3¡,3* and *PyuARF34*) appeared highly expressed in the stem, indicating that these *ARFs* are important for stem development, such as xylem development. In this study, the lignin content of cuttings and seedlings was measured under light treatment. The lignin content was lower, and the gene transcription profiles were more limited in red light than in white light on days 1, 7, and 14, suggesting that *PyuARF16/33* might be a good candidate gene regulating lignin synthesis and the rapid growth of poplar cuttings and seedlings under red light.

Previous studies have shown that hierarchical transcriptional regulatory networks composed of various transcription factors such as *NACs* and *MYBs* are responsible for regulating lignin biosynthesis in *A. thaliana* [46,47,48]. For example, *PbrMYB169* positively regulates the lignification of stone cells in pear fruit [49], and *OSNAC5* regulates drought tolerance and lignin accumulation in roots [50]. A recent study showed that *ARF8*–*MYB26*–*NST1* modulates premature endothecium lignification in *A. thaliana* by regulating lignin synthesis [51]. This study predicts that *PyuARF16/33* might also regulate lignin synthesis through the downstream genes *MYBs*. However, whether *PyuARF16/33* affects lignin accumulation by regulating *MYB26*–*NST1* needs further experimental verification.

## 5. Conclusions

We identified 35 *ARF* family genes in *P. yunnanensis* genome and verified the tissue specificity of 12 *PyuARFs* with transcriptional activation function. We then selected five *PyuARFs* for transcription profiles analysis under light treatment. Finally, we measured lignin content under light treatment. The results showed that the lignin content was lower, and the gene transcription profiles were more limited in red light than in white light on days 1, 7, and 14, suggesting that *PyuARF16/33* might be a good candidate gene regulating lignin synthesis and the rapid growth of cuttings and seedlings under red light in *P. yunnanensis.*

## Figures and Tables

**Figure 1 genes-14-00278-f001:**
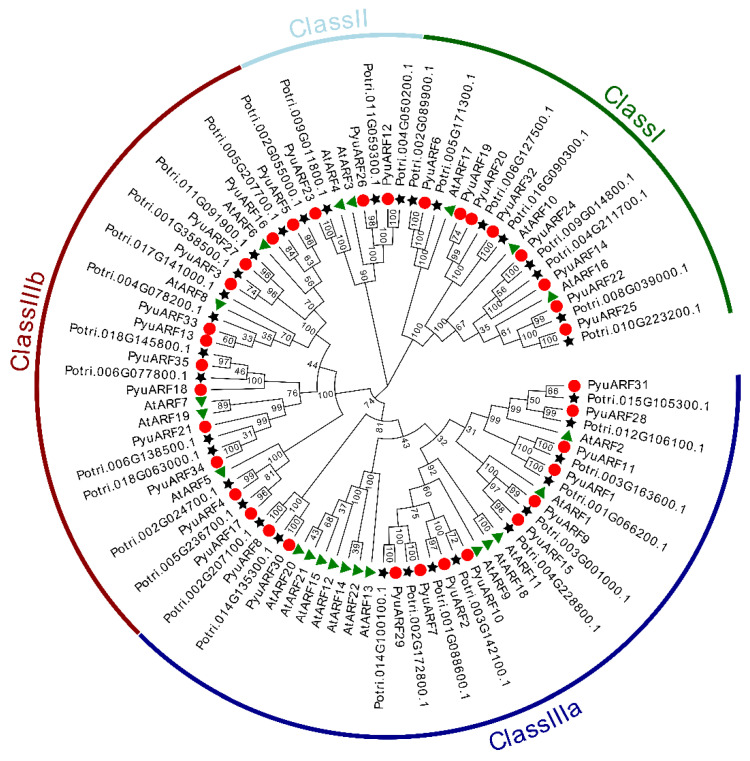
Phylogenetic tree comparison of the *ARFs* from *P. trichocarpa*, *P. yunnanensis* and *A. thaliana*. We constructed a rootless phylogenetic tree using the protein sequences of the *ARF* family members, i.e., 22 *AtARFs* from *A. thaliana*, 35 *PtriARFs* from *P. trichocarpa*, and 35 *PyuARFs* from *P. yunnanensis.* The red circles represent the genes of *P. yunnanensis*, the green triangles represent the genes of *A. thaliana*, and the black five-pointed stars represent the genes of *P. trichocarpa*.

**Figure 2 genes-14-00278-f002:**
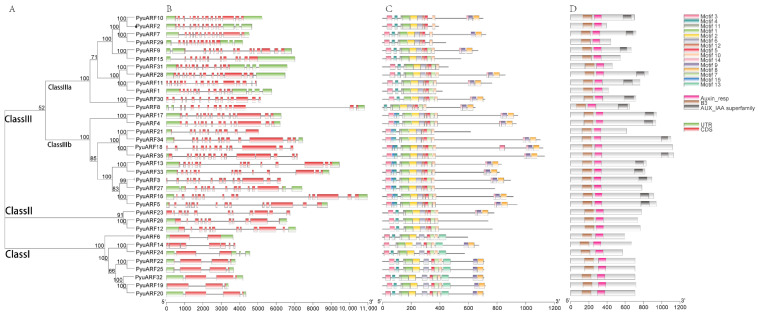
Analysis of the conserved *ARF* DNA-binding domains, gene structure, and conserved motifs depending on the phylogenetic relationships in *ARF* genes. A: we built the phylogenetic tree using 35 ARF proteins with the ML method. The phylogenetic tree contains three subgroups, Class I, Class II, Class III (Class IIIa, Class IIIb), shown in different colors. B: Exon/intron structure analysis of the *ARF* genes. Black lines, red boxes, and green boxes indicate introns, exons, and untranslated regions (UTRs), respectively. C: Conserved motifs of the *ARF* genes elucidated by Multiple Expectation maximizations for Motif Elicitation (MEME). The conserved motifs are represented by the different colored boxes. D: we examined the conserved *ARF* domains using Pfam and the Simple Modular Architecture Research Tool (SMART). Brown boxes indicate conserved *ARF* DNA-binding domains, blue boxes indicate conserved *ARF* domains, and gray boxes indicate conserved *AUX/IAA* domains. The scale bar of each *PyuARFs* is shown at the bottom of the Figure.

**Figure 3 genes-14-00278-f003:**
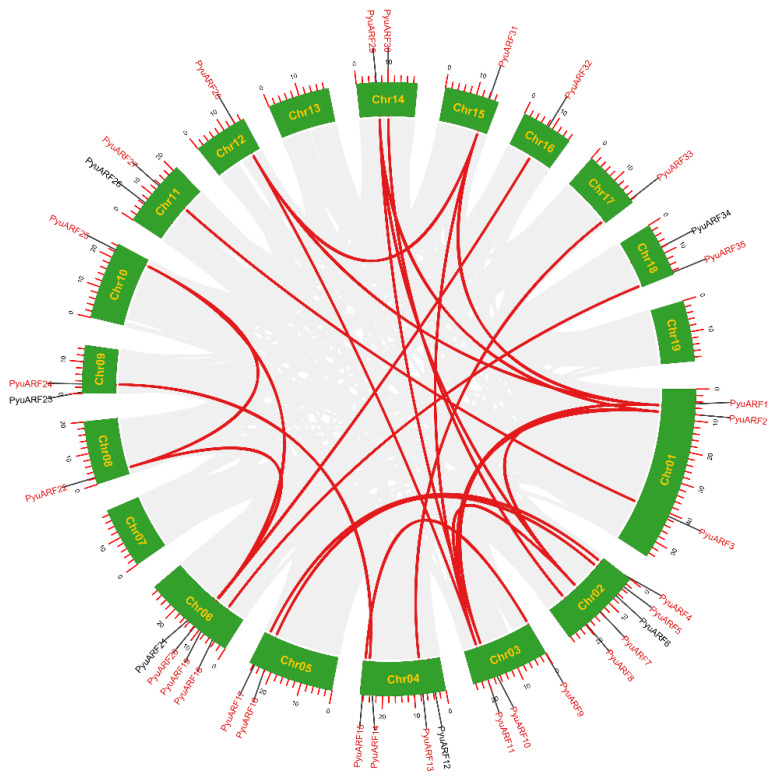
Synteny analysis of *PyuARFs*. The genes connected by red lines indicate *ARFs* that are homologous among the *PyuARFs*, and the genes connected by gray lines indicate other homologous genes.

**Figure 4 genes-14-00278-f004:**
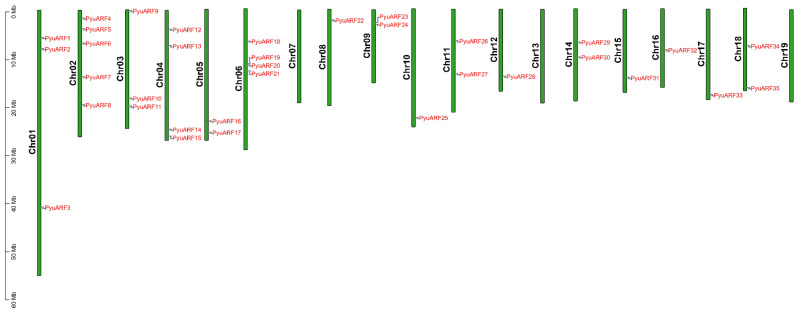
Schematic representation of the chromosomal distribution of the *PyuARFs*. The vertical bars indicate the chromosomes of *P. yunnanensis*. The chromosome number is located to the left of each chromosome. The scale on the left represents the chromosome length. The *PyuARF* genes are marked in red.

**Figure 5 genes-14-00278-f005:**
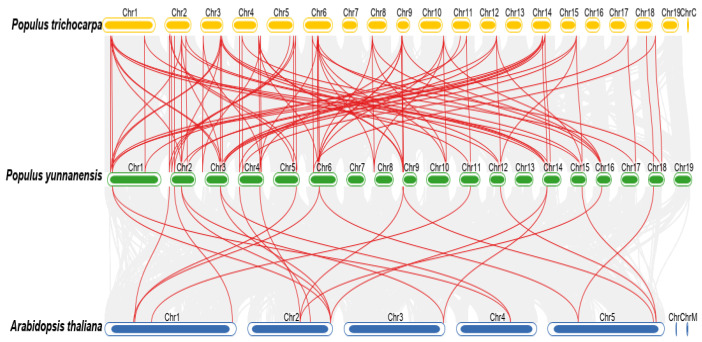
Collinearity analysis of *ARFs* between *P. trichocarpa*, *P. yunnanensis*, and *A. thaliana*. Chr represents chromosome-scale scaffolds. The putative collinear genes in *P. yunnanensis* and the other two species are marked in gray, while the syntonic *ARF* gene pairs are marked in red.

**Figure 6 genes-14-00278-f006:**
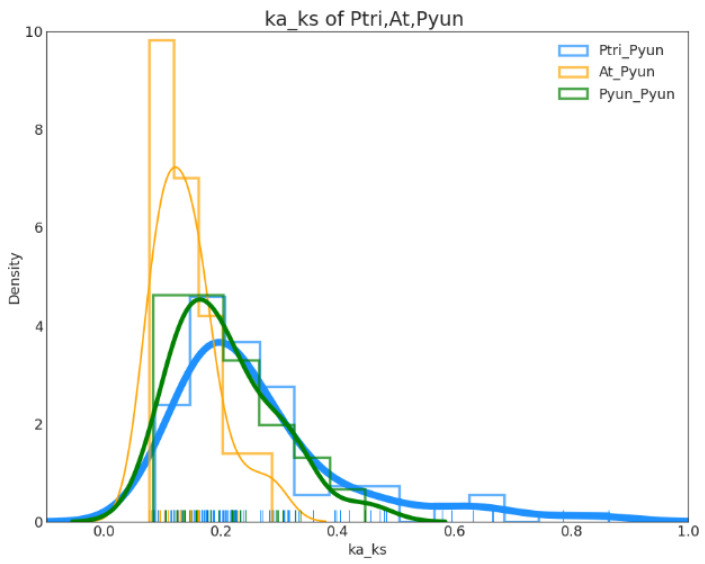
Ka/Ks ratios of collinear *PyuARFs* in *P. trichocarpa*, *P. yunnanensis*, and *A. thaliana*. Blue curves represent the Ka/Ks ratios of collinear *ARF* genes in *P. trichocarpa* and *P. yunnanensis*. Yellow curves represent the Ka/Ks ratios of collinear *ARF* genes in *P. yunnanensis* and *A. thaliana*. Green curves represent the Ka/Ks ratios of collinear *ARF* genes in *P. yunnanensis* and *P. yunnanensis*.

**Figure 7 genes-14-00278-f007:**
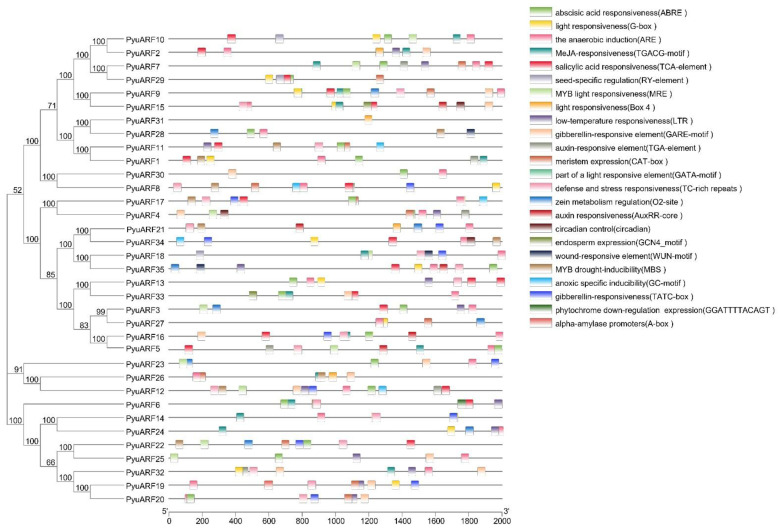
Cis-acting elements in the promoter regions of 35 *PyuARFs*. The black line indicates the promoter length of the *PyuARFs*. The different colored boxes on the right represent cis-acting elements with different functions.

**Figure 8 genes-14-00278-f008:**
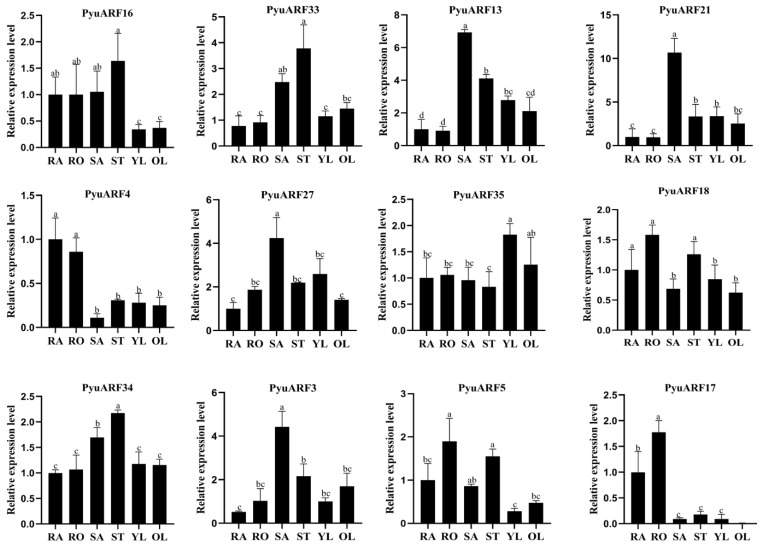
Tissue-specific transcription profiles of 12 *PyuARFs* with transcriptional activation function. RAM: root apical meristem, Root: mature heel, SAM: shoot apical meristem, Stem: mature Stem, YL: young leaf, OL: mature leaf. The ordinate represents the transcription profiles of the genes in these tissues. Data represent mean ± SD from at least three independent experiments. The same letter indicates a significant correlation at the 0.05 probability level, while different letters indicate no difference.

**Figure 9 genes-14-00278-f009:**
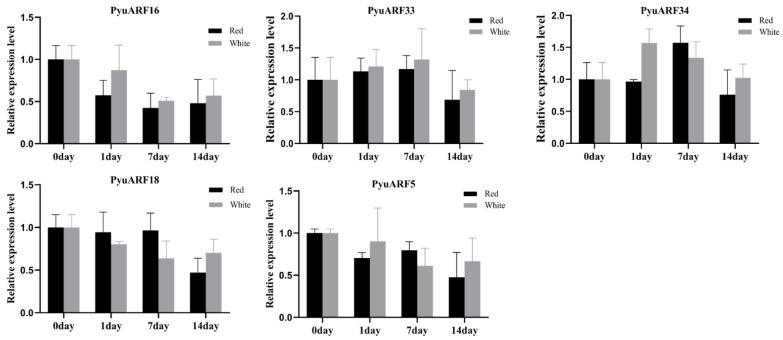
Transcription profiles of 5 *PyuARFs* with high expression in the stem under light. The abscissa represents the days of treatment, and the ordinate represents the transcription profiles of the genes.

**Figure 10 genes-14-00278-f010:**
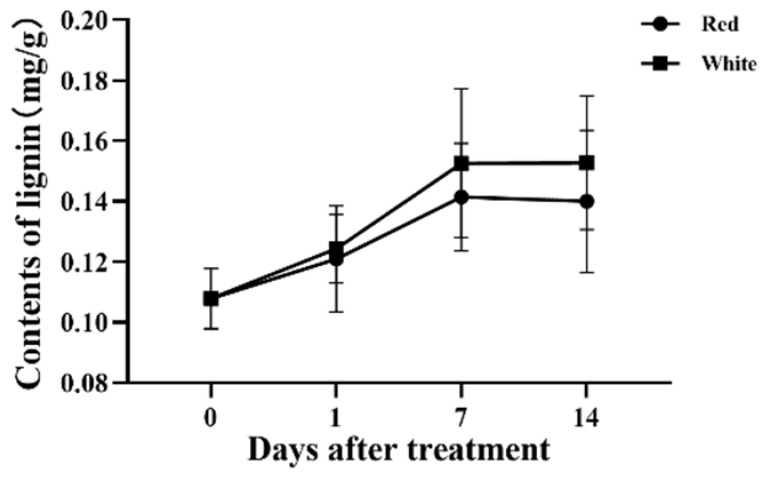
Lignin content in the stem of *P. yunnanensis* on days 1, 7, and 14 under light irradiation. The abscissa represents the days of treatment, and the ordinate represents the content of lignin.

**Figure 11 genes-14-00278-f011:**
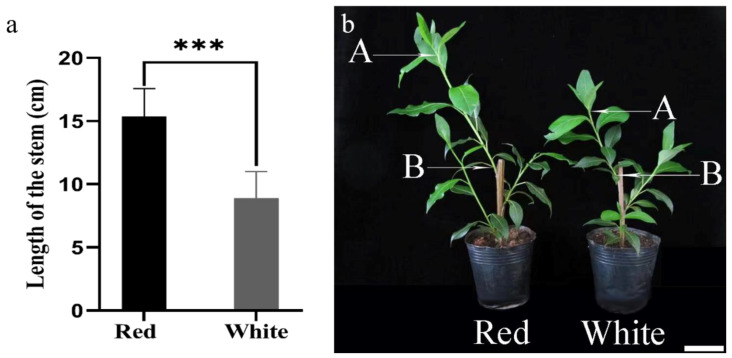
(**a**): Heights of the stems after 14 days of growth under white and red light. (**b**): Plants grown for 14 days under white and red light. ***: Significant correlation at the 0.001 probability level. Error bars indicate the SD of three independent biological and technical replicates. Bar = 5 cm.

**Table 1 genes-14-00278-t001:** Characteristics of the identified *PyuARF*s family members in *P. yunnanensis*.

Gene ID	Genomic Accession	Location	Extron	Fragment Length	Molecular Weight (kDa)	Amino Acid(aa)	Isoelectric Point (pI)
*PyuARF1*	*Pyu01G006120*	5736573–5742324	15	1251	46.78	416	8.1
*PyuARF2*	*Pyu01G008450*	8031542–8036207	14	1176	43.46	391	7.67
*PyuARF3*	*Pyu01G034670*	41065699–41071947	14	2682	98.8	893	6.32
*PyuARF4*	*Pyu02G002280*	1660312–1666527	14	2802	102.01	933	5.53
*PyuARF5*	*Pyu02G005050*	3879016–3887810	14	2817	103.59	938	6.06
*PyuARF6*	*Pyu02G008360*	6845869–6850647	2	1785	65.65	594	5.77
*PyuARF7*	*Pyu02G015810*	13869784–13874301	13	2163	80.26	720	6.39
*PyuARF8*	*Pyu02G019930*	19706536–19717355	14	1941	71.54	646	6.34
*PyuARF9*	*Pyu03G000090*	171315–178144	15	1998	73.94	665	5.9
*PyuARF10*	*Pyu03G013380*	18392114–18397334	14	2106	77.69	701	6.23
*PyuARF11*	*Pyu03G015300*	20036875–20041785	16	2289	86.08	762	6.4
*PyuARF12*	*Pyu04G003960*	3912233–3919295	12	2298	83.72	765	6.65
*PyuARF13*	*Pyu04G006900*	7350686–7360129	13	2493	92.51	830	5.93
*PyuARF14*	*Pyu04G020520*	24752857–24756606	7	2016	74.07	671	8.83
*PyuARF15*	*Pyu04G021840*	26238695–26245709	13	1641	60.74	546	6.58
*PyuARF16*	*Pyu05G019310*	23213022–23223997	15	2739	100.59	912	6.11
*PyuARF17*	*Pyu05G022180*	25600702–25606965	14	2835	103.59	944	5.28
*PyuARF18*	*Pyu06G007120*	6766693–6773609	14	3360	123.67	1119	6.28
*PyuARF19*	*Pyu06G011650*	11498093–11502446	3	2151	79.01	716	6.61
*PyuARF20*	*Pyu06G011820*	11768329–11772667	4	2118	77.62	705	6.55
*PyuARF21*	*Pyu06G012840*	13008364–13013396	11	1842	69.57	613	8.16
*PyuARF22*	*Pyu08G003290*	2235865–2239624	4	2127	78.33	708	6.82
*PyuARF23*	*Pyu09G001470*	2631312 2638053	12	2337	86.28	778	6.55
*PyuARF24*	*Pyu09G001720*	3039240–3043796	5	1719	63.81	572	8.99
*PyuARF25*	*Pyu10G020920*	22638122–22641788	4	2121	77.67	706	7.55
*PyuARF26*	*Pyu11G005450*	6563914–6570489	11	2211	80.79	736	6.51
*PyuARF27*	*Pyu11G008720*	13448442–13455848	15	2349	86.35	782	6.43
*PyuARF28*	*Pyu12G009540*	13934942–13941420	15	2571	95.14	856	6.3
*PyuARF29*	*Pyu14G007780*	6952860–6957024	14	1326	49.41	441	7.3
*PyuARF30*	*Pyu14G011450*	10055393–10060539	14	2145	80.07	714	6.84
*PyuARF31*	*Pyu15G010310*	14206705–14213300	15	1377	51.26	458	8.95
*PyuARF32*	*Pyu16G008410*	8570402–8574578	4	2121	78.49	706	8.27
*PyuARF33*	*Pyu17G014300*	17824825–17833704	14	2448	91.21	815	6.06
*PyuARF34*	*Pyu18G005570*	7869669–7877113	14	3303	123.61	1100	5.74
*PyuARF35*	*Pyu18G012790*	16594493–16602516	15	3393	125.45	1130	6.37

## Data Availability

Genomic data of *P. yunnanensis* can be obtained by contacting the corresponding author. The data that support the findings of this study are available from the corresponding author upon reasonable request.

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
