# Peer review of "PyuARF16/33 Are Involved in the Regulation of Lignin Synthesis and Rapid Growth in Populus yunnanensis"

_genes, 2023, doi:10.3390/genes14020278_

Round 1

Reviewer 1 Report

Dear Authors,

In general, the manuscript is interesting. However it need some modifications, corrections and additions. 

In the Introduction section, the aim of the study must be clearly indicated. In the present form, it gets lost easily .

Names of genes should be italicized but the names of proteins shouldn't be.

All Latin names should be in italics. The abbreviation of the species name must include a dot.

Table 1. There should be a reference to the genome assembly in the table description.

Line 193: The line division mark is missing "... enviroment [34] 2.2 Identifi-"

Figure 1. There is no explanation of what the symbols mean i.e. circle, triangle and star.

Lines 213 - 232: This fragment do not contain results. It contains background information. It should be removed and can be placed in the introduction or in the dissussion section. 

Lines 272- 274: as above. Move it to the discussion or the conclusion section. 

In general, in a more detailed and synthetic way.

Lines 334 - 336: How did you determine the consistency of lignin content and PyuARF33/16 expression?

The discussion section needs to be expanded. For many results sections, the discussion is missing. 

Best,

R.

ults should be decribed in more detail

Reviewer 2 Report

In this work, Zhixu Hu and coworkers identified putative 35 ARFs family genes from P. yunnanensis genome in order to characterize their chromosome positions, functional features for their genes, proteins and promoters, their tissue-specific expression and expression profile response to the red light, as well as determination of temporal accumulation of lignin content, determination of temporal accumulation of lignin content as well as to identify the best candidate gene(s) potentially involved in in the synthesis of secondary cell wall. It should be noted that auxin are involved in plant development including embryo differentiation, vascular tissue differentiation, microtubule differentiation, root and stem morphogenesis, fruit ripening, apical dominance, phototropism plant flowering and diurnal opening and closure, flower opening and development  of secondary xylem and are characterized for a few plants, but the molecular mechanisms of their participation in these cellular processes of different plant species require clarification.

Initially, researchers searched for candidate genes for ARFs genes in the P. yunnanensis genome. Further phylogenetic and feature domain analysis showed that (i) ARF family genes were mainly divided into three categories (Class I, Class II and Class III); (ii) Class III mainly included two subcategories, IIIa and IIIb; and (iii) MR domain of the ARF gene family of Class IIIb mainly contains Q (glutamine), which is presumed to have transcriptional activation activity, MR domain of the ARF genes of class IIIb mainly contains SPL (serine, proline and leucine) which is presumed to have transcriptional inhibition activity. Further, the researchers conducted a search for cis-acting elements in ARF gene promoters using the PLACE website in order to identify cis-acting elements associated with the hormone response in plants. This in silico analysis revealed that the majority of the ARF promoters contained regulatory elements that were sensitive to light, plant hormones and stress.

Several experiments were performed to establish of the tissue-specific expression profile of ARF genes, estimated the transcriptional dynamics of ARF genes under red light irradiation as well as find out the relationship between the level of transcription of the ARF of genes and the content of lignin in experiments with different processing of light.

Major conclusions by the authors include (i) the 35 ARFs genes were identified in P. yunnanensis genome; (ii) ARF promoters contained regulatory elements that were sensitive to light, plant hormones and stress; (iii) the some ARF have preferential expression patterns; (iv) the tissue-preferential and stage-specific expressions exhibited by several ARF genes in stem; (v) PyuARF16/33 were are a suitable candidate genes to regulate the change of lignin content.

The topic of this work is interesting; however, there are some comments to manuscript:

Major comments:

1.    When the authors describe the results of the expression profiles of PyuARF genes at different organs, and light treatment readers may find it interesting whether there are dependencies between the presence of regulatory cis-elements in PyuARF promoter regions and their functional role in organ-specific expression and light treatment?

2.    It is not clear on the basis of what considerations 12 PyuARF genes of 35 were selected to assess the tissue-specific level of their transcription?

3.    In relation to the results in a comparative assessment of changes in transcription profile and the conclusion of the authors that «The lignin content was consistent with the changes of the expression profiles of PyuARF33/16 » (Figures 10 and 11). Based on what changes in the level of transcription, such a conclusion is made. The level of lignin as processing and white and red-light increases, and the transcription level Pyuarf33/16 changes differently: for PyuARF33 of reliable changes up to 7 days of processing is not marked, and on the 14th day the level of transcription is reduced; for PyuARF16 - the level of transcription by 7 days is reliably reduced. Should clearly describe how the authors came to the conclusion that « the lignin and PyuARF33/16 might play crucial roles in stem during P. yunnanensis of fast- growing upon red light irradiation», since this is a crucial manuscript conclusion.

Minor comments:

1.    Line 136 – Add еhe name of the device - ultra-micro spectrophotometer K5800C.

2.    Line 188 – “The protein contains 2 (PyuARF6) to 16 (PyuARF11) exons in genes” – Proteins cannot contain intron, so intron can only be in the genes that encode proteins. Correction should be made.

3.    Lines 193-194. What does it mean “2.2 Identification and sequence analysis of ARFs in P. yunnanensis”?

4.    It is more professional to use the term “5’-terminus” instead of “5’-end”, as well as the term “3’-terminus” instead of “3’-end”. Check by the manuscript text.

5.    In the figures 8 and 9  - Relative expression levels, the same axes should be represented - Relative Fold Difference or Relative Fold Difference (2-ΔCt). It is more professional to use the term “Transcription profiles…’ instead of “Expression profiles….”.

Round 2

Reviewer 1 Report

Dear Authors,

You have made some of the changes suggested in the previous review. Unfortunately, not all of them.  I have the following comments on this manuscript:

Line 53 a full name of the species with authority is necessary.

Line 119 and many subsequent lines is A thaliana and should be A. thaliana.

Many added passages have grammatical and stylistic errors.

Is the genome for P. yunnanensis in open access? 

The discussion still needs a significant amount of work. E.g., fragments from lines 367-379 need to be placed in the background of the literature data. 

Best,

R.

Reviewer 2 Report

After reviewing the resubmitted manuscript and the authors' responses to my comments, the following should be noted: (i) the authors satisfactorily responded to all the questions and comments I have raised; (ii) the comments and questions posed by me has been clarified in manuscript. However, there are comments:

Major comment:

1.    In relation to the results in a comparative assessment of changes in transcription profile and the conclusion of the authors that «Both lignin content and gene transcription profiles were lower in red light than in white light at day 1, 7, and 14, suggesting that PyuARF16\33 might be a good candidate gene for lignin synthesis to regulate the rapid growth of poplar cutting seedlings under red light” (Figure 9 and 10). From the response to the reviewer's comments on this question, as well as from the text of the manuscript, it is not clear whether the authors found statistically significant differences in the levels of transcription of the target PyuARF16\33 genes and in the amount of lignin under red light compared to white light? Judging by the presented data, no statistically significant change in these parameters was found in plants grown under different light conditions. In my opinion, we can only talk about a tendency to a decrease in both the level of transcription of the studied genes and lignin, depending on the light conditions. For the conclusion, it should be proved that the differences in these indicators are statistically significant, as well as indicate how the reliability of the identified differences was assessed. Such data and explanations are extremely important since this is the main conclusion of the authors of the manuscript.

Minor comment:

1.            Designation of genes and proteins - as a rule, the name of the genes is marked in italics, in contrast to proteins, which are not marked in italics. In particular, section 3.1 Genome-wide Identification and Sequence Analysis of PyuARFs, both when characterizing protein products and gene sequences, the authors marked all names in italics.
